# Genetic Load of Alternations of Transcription Factor Genes in Non-Syndromic Deafness and the Associated Clinical Phenotypes: Experience from Two Tertiary Referral Centers

**DOI:** 10.3390/biomedicines10092125

**Published:** 2022-08-30

**Authors:** Hyung Dong Jo, Jin Hee Han, So Min Lee, Dong Hwa Choi, Sang-Yeon Lee, Byung Yoon Choi

**Affiliations:** 1Department of Otorhinolaryngology-Head and Neck Surgery, Seoul National University Hospital, Seoul National University College of Medicine, Seoul 03080, Korea; 2Department of Otorhinolaryngology-Head and Neck Surgery, Seoul National University Bundang Hospital, Seoul National University College of Medicine, Seongnam 13620, Korea; 3Sensory Organ Research Institute, Seoul National University Medical Research Center, Seoul 03080, Korea

**Keywords:** transcription factor variants (TF), non-syndromic hearing loss, *POU3F4*, *POU4F3*, *LMX1A*, *EYA4*

## Abstract

Sensorineural hearing loss is one of the most common inherited sensory disorders. Functional classifications of deafness genes have shed light on genotype- and mechanism-based pharmacological approaches and on gene therapy strategies. In this study, we characterized the clinical phenotypes and genotypes of non-syndromic deafness caused by transcription factor (TF) gene variants, one of the functional classifications of genetic hearing loss. Of 1280 probands whose genomic DNA was subjected to molecular genetic testing, TF genes were responsible for hearing loss in 2.6%. Thirty-three pathogenic variants, including nine novel variants, accounting for non-syndromic deafness were clustered in only four TF genes (*POU3F4*, *POU4F3*, *LMX1A*, and *EYA4*), which is indicative of a narrow molecular etiologic spectrum of TF genes, and the functional redundancy of many other TF genes, in the context of non-syndromic deafness. The audiological and radiological characteristics associated with the four TF genes differed significantly, with a wide phenotypic spectrum. The results of this study reveal the genetic load of TF gene alterations among a cohort with non-syndromic hearing loss. Additionally, we have further refined the clinical profiles associated with TF gene variants as a basis for a personalized, genetically tailored approach to audiological rehabilitation.

## 1. Introduction

Congenital hearing loss affects 1–2 out of every 1000 newborns and is one of the most common inherited sensory disorders in humans [1]. Genetic causes account for approximately 50% of sensorineural hearing loss (SNHL) cases, with recent advances in genomics contributing to the identification of more than 150 genes implicated in its etiology (https://hereditaryhearingloss.org/, accessed on 1 February 2022) [2]. Genetically engineered in vivo and in vitro models were used to establish functional classifications of deafness-related genes, including (1) hair bundle development and functioning; (2) synaptic transmission; (3) cellular adhesion and maintenance; (4) cochlear ion homeostasis; (5) extracellular matrix; (6) oxidative stress, metabolism, and mitochondrial defects; and (7) transcriptional regulation [3]. The application of these functional assignments provides a better understanding of genetic hearing loss, including mechanism- and genotype-based pharmacological and gene therapy approaches.

The “central dogma” refers to the transfer of sequence information between RNA, DNA, and proteins within a biological system. It describes how the information embedded in DNA is transferred to mRNA (transcription) and how amino-acid chains are synthesized from mRNA (translation) [4]. Transcription factors (TFs) recognize specific DNA sequences to control transcription by forming a complex that guides genome expression [5]. TFs generally contain several domains (effector, DNA-binding, and regulatory domains) that regulate their localization, chromatin accessibility, and transcriptional activity. More than 1600 human TFs have been documented in the literature [5], and their variants have been implicated in diverse diseases and syndromes, including cardiovascular diseases, cancer, neurological disorders, autoimmune diseases, and diabetes [6]. However, only a handful of TF variants are known to cause hearing impairment, and their clinical phenotypes and genotypes in the context of hearing loss remain poorly understood. The delineation of audiological phenotypes related to TF genes may be clinically valuable, shedding light on the mechanism of SNHL and the potential for timely intervention.

In this study, we characterized the audiological phenotypes and genotypes of disease-causing TF variants, as one of the functional classifications of genetic hearing loss (transcriptional regulation), with the aim of revealing genotype-phenotype correlations. Our results expand the genetic spectrum of TF variants associated with non-syndromic deafness and further refine the associated clinical profiles.

## 2. Materials and Methods

### 2.1. Subjects

All procedures in this study were approved by the Institutional Review Boards of Seoul National University Hospital (IRB-H-0905-041-281) and Seoul National University Bundang Hospital (IRB-B-1007-105-402). Written informed consent was obtained from all participants or the legal guardians of the pediatric participants. The study consisted of a retrospective review using the in-house databases of genetic hearing loss from the two participating hospitals. The DNA of 1280 probands was subjected to molecular genetic testing regardless of any specific audiologic phenotypes or modes of inheritance. Families with genetically confirmed disease-causing TF variants were included (*n* = 55, 4.3%). Families harboring genetic variants implicated in syndromic deafness (*n* = 22, 1.7%), primarily Waardenburg syndrome and branchio-oto-renal syndrome, were excluded. Ultimately, 33 families (2.6%) with TF-associated non-syndromic deafness were included.

### 2.2. Clinical Phenotyping

Comprehensive phenotypic evaluations, including audiological imaging, were performed in all patients and the postoperative outcome of cochlear implantation (CI) was documented. Audiological evaluations consisted of pure tone audiometry (PTA) in adults, and the auditory brainstem response threshold (ABRT) in prelingual infants. Hearing levels (HLs) were determined by averaging the hearing thresholds at 0.5, 1, 2, and 4 kHz of the PTA and were categorized as mild (25–40 dB HL), moderate (41–55 dB HL), moderately severe (56–70 dB HL), severe (71–90 dB HL), or profound (>91 dB HL) [7,8]. The audiograms were classified into four categories according to their configuration: flat, down-sloping, rising, or U-shaped [9,10]. Audiograms were defined as flat if the thresholds across frequencies did not differ from each other by >20 dB. Down-sloping audiograms were those with thresholds that occurred at equal or successively higher levels from 0.25 kHz to 8 kHz, with the difference between thresholds at 0.25 kHz and 8 kHz being >20 dB. Rising audiograms were defined as having thresholds that occurred at equal or successively lower levels from 0.25 kHz to 8 kHz, with the difference between thresholds at 0.25 kHz and 8 kHz being >20 dB. Audiograms were defined as U-shaped if one or more adjacent thresholds between 0.5 kHz and 4 kHz was >20 dB compared to the poorer threshold of 0.25 kHz or 8 kHz. Asymmetric hearing loss was defined as an interaural difference >10 dB in the threshold average of 0.5, 1, 2, and 4 kHz, with the HL of the better-hearing ear being worse than 25 dB HL [9,11]. Radiological evaluation of the middle ear, inner ear, and internal acoustic canal (IAC) included temporal bone computed tomography (CT) and internal acoustic canal magnetic resonance imaging (IAC MRI). Anatomical abnormalities of the middle ear, inner ear, and IAC were evaluated and classified using Sennaroglu’s revised classifications of cochleovestibular malformations [12]. The audiologic performance outcome of each cochlear implantee was evaluated by comparing the Categories of Auditory Perception (CAP), the Korean version of the Central Institute for the Deaf Everyday Speech Sentence Test (K-CID), Phonetically Balanced Word (PB), and spondee scores based on the speech evaluations conducted before and after CI. The postoperative speech evaluations were performed 3, 6, 12, 18, and 24 months after CI [13].

### 2.3. Molecular Genetic Testing

Genomic DNA was extracted from peripheral blood using a standard procedure. Initial screening was usually conducted with real-time PCR mutational hotspot screening kits, which targeted 22 variants of 10 hearing loss genes (*GJB2*, *SLC26A4*, *CDH23*, *TMPRSS3*, *MT-RNR1*, *OTOF*, *MPZL2*, *TMC1*, *COCH*, and *ATP1A3*), based on the prevalence of the genetic diagnostic yield in Korea and genotype–phenotype correlation (e.g., auditory neuropathy) [14,15]. Specifically, for a particular phenotype, direct sequencing was performed preferentially, followed by targeted panel sequencing (Otogenetics, Norcross, GA, USA) or whole-exome sequencing. The resulting reads were compared with the UCSC hg19 reference genome, and non-synonymous single-nucleotide polymorphisms (SNPs) were filtered at a depth of 40; dbSNP138 was filtered out, except for the flagged SNP. Disease-causing variants were detected in this study using a comprehensive filtering process and bioinformatics analyses, as described previously [16,17,18]. The pathogenic potential of the novel variants identified herein was evaluated according to the ACMG/AMP guidelines developed for hearing loss [19].

### 2.4. Statistical Analysis

Descriptive statistical analyses of the postoperative speech evaluation scores were performed using SPSS for Windows, version 18.0 (IBM Corp., Armonk, NY, USA). The schematic illustrations were created using GraphPad Prism version 9.0.3 for Windows (GraphPad Software LLC, San Diego, CA, USA; www.graphpad.com, accessed on 1 February 2022). The Mann–Whitney U test was used to compare postoperative CI outcomes between groups. Statistical significance was defined as a *p*-value < 0.05.

## 3. Results

### 3.1. Distribution of TF Genes

Among 720 probands in whom causative deafness variants were identified, 33 were from families with TF-associated non-syndromic deafness. The causative TF genes of these families were *POU3F4* (*n* = 16, 48.5%), *POU4F3* (*n* = 6, 18.2%), *LMX1A* (*n* = 6, 18.2%), and *EYA4* (*n* = 5, 15.2%) (Table 1). The clustering of the deafness-causing TF variants in only four genes highlighted their role in non-syndromic deafness in Korea. Among the 33 variants detected in total, nine were novel: *POU3F4* (c.458delC:p.Pro153Leufs*88, c.989G>A:p.Arg330Lys, c.958G>T:p.Glu320*, and c.626A>G:p.Gln229Arg), *LMX1A* (c.331del:p.Gln111Argfs*7), and *EYA4* (c.697C>T:p.Gln233*, c.208+1del, c.578dup:p.Tyr193*, and c.1468G>T:p.Glu490*) (Table 2). All nine novel TF gene variants satisfied the criteria of the ACMG/AMP guidelines defining pathogenic or likely pathogenic genes (Table 2).

### 3.2. POU3F4

#### 3.2.1. *POU3F4*: Genotype Profile

*POU3F4* variants, segregating as an X-linked or de novo trait, were detected in 18 patients from 16 families. Most variants resided within the coding region of *POU3F4*, but four were copy number variants consisting of a large genomic deletion in the DFNX2 locus (Figure 1a). Large genomic deletions located upstream of *POU3F4* were previously reported and presumably affect gene expression by disturbing the promotor or enhancer. Moreover, four of the 16 families harbored novel hemizygous *POU3F4* variants (c.458delC:p.Pro153Leufs*88, c.989G>A:p.Arg330Lys, c.958G>T:p.Glu320*, and c.626A>G:p.Gln229Arg).

#### 3.2.2. *POU3F4*: Audiological Profile and Cochlear Implantation Results

Of the 18 patients, 10 (55.6%) had MHL and 8 (44.4%) severe-to-profound SNHL. Hearing loss progression was documented in 40% of MHL patients, whereas hearing loss in all SNHL patients was non-progressive, due to a severe-to-profound loss already at baseline. As expected, temporal bone CT revealed incomplete partition (IP) type III in all patients. Eleven (61.1%) of these patients had undergone unilateral or bilateral CI, with a mean age at implantation of 6.9 (SD: 26.6) years. In the 11 patients (13 ears), the average CAP score was 1.3 (SD: 1.9) at baseline, 2.6 (SD: 1.6) at 3 months postoperatively, 3.0 (SD: 1.6) at 6 months postoperatively, 3.4 (SD: 1.2) at 12 months postoperatively, 3.7 (SD: 1.5) at 18 months postoperatively, and 3.8 (SD: 1.3) at 24 months postoperatively. Among the 13 ears, only one (SH65) had a score > 6 after 2 years: in the other ears it remained at or below 5 (i.e., understanding common phrases but not sentences without lip-reading). The postoperative CI outcomes were also compared with an age-, sex-, and laterality-matched cohort of *GJB2*-associated cochlear implantees (control group) (Table 3). Although the CAP scores at baseline, 3 months, and 6 months post-CI were not significantly different, the postoperative CAP scores of *POU3F4* patients were significantly and progressively poorer than those of the controls at 12, 18, and 24 months after implantation (*p* < 0.05). The determination of postoperative CI outcomes depending on the genotype (missense or C-terminal extension variants versus truncated or copy number variants) did not reveal a definite genotype–phenotype correlation (Figure 2a).

### 3.3. POU4F3

#### 3.3.1. *POU4F3*: Genotype Profile

Ten patients from six *POU4F3*-associated families were identified, and in all of them an autosomal dominant inheritance pattern was determined. All familial variants were missense or frameshift variants within the two functional domains, including the POU-specific and POU homeodomain (Figure 1b). Notably, the variant in SB218 was a truncated variant residing in the POU-specific domain, thus affecting the two nuclear localization signal (NLS) domains within the POU homeodomain.

#### 3.3.2. *POU4F3*: Audiological Profile and Cochlear Implantation Results

Nine of the ten patients (90%) in the *POU4F3* group had SNHL, except one patient who had mixed hearing loss. The audiograms had a U-shaped configuration, characterized by a mid-frequency notch at 1–2 kHz in five patients (50.0%). Down-sloping (*n* = 3, 30.0%), mixed hearing loss (*n* = 1, 10.0%), and flat (*n* = 1, 10.0%) configurations characterized the audiograms of the remaining patients. The severity of hearing loss tended to be moderate to moderately severe initially but progressed thereafter. In three patients, their hearing loss eventually deteriorated to severe-to-profound, and they underwent CI at a mean age of 41.3 years (SD: 13.1). One patient was implanted bilaterally in a single procedure, and the other two patients were implanted unilaterally. The CI outcomes were favorable, with K-CID, PB, and spondee scores above 96%, 70%, and 70% at the 1 year postoperative exam, respectively (Figure 2b). One patient (SB216) displayed bilateral moderate SNHL in her early 30s and opted for bilateral middle ear implantation (MEI) surgery rather than a hearing aid due to unsatisfactory experience with conventional hearing aids. She has been a satisfied user of MEI for 6 years.

### 3.4. LMX1A

#### 3.4.1. *LMX1A*: Genotype Profile

Nine patients from six *LMX1A*-associated families were identified. In most cases, the pedigrees indicated an autosomal dominant inheritance pattern. In one family, a de novo heterozygous missense variant (c.595A>G:p.Arg199Gly) was previously reported [18]. Four of the six variants were in the homeodomain, and the remaining two were truncated variants in LIM2 and the C-terminus, respectively (Figure 1c). Of these, a novel heterozygous *LMX1A* variant (c.331del:p.Gln111Argfs*7) was identified.

#### 3.4.2. *LMX1A*: Audiological Profile and Cochlear Implantation Results

Asymmetric hearing loss (interaural difference > 15dB) was identified in five of the nine patients. The average asymmetry between the two ears was 35.75 dB (range 15–65). Three of the four *LMX1A* patients who were eligible for follow-up audiometry reported progressive hearing loss. In one patient (SB727), hearing deteriorated to profound hearing loss in her left ear and the asymmetric hearing loss remained. The patient eventually underwent unilateral CI, with significant improvement in her speech perception scores 3 and 6 months postoperatively (Figure 2c). One patient (SH407) reported hearing fluctuations in the left ear, with intermittent episodes of vertigo and headache, likely indicating Meniere’s disease.

### 3.5. EYA4

#### 3.5.1. *EYA4*: Genotype Profile

In the eight patients from five *EYA4*-associated families, an autosomal dominant pattern was determined. Of the five identified variants, two nonsense and one deletion variant were localized within the variable region (eya-VR), and two loss-of-function variants, one frameshift variant, and one nonsense variant in the homologous domain (eya-HR) (Figure 1d). Of these, four novel heterozygous *EYA4* variants (c.697C>T:p.Gln233*, c.208+1del, c.578dup:p.Tyr193*, and c.1468G>T:p.Glu490*) were identified.

#### 3.5.2. *EYA4*: Audiological Profile and Cochlear Implantation Results

Among the eight patients, six (75.0%) had SNHL and two (25.0%) MHL, with postlingual onset (average age of onset: 32.5 ± 14.9 years). The severity of hearing loss ranged from moderate to severe. A down-sloping audiogram was the most prevalent configuration (five patients, 62.5%). The audiometric configuration deteriorated gradually in seven out of eight patients (87.5%). None of the patients reported symptoms of vestibulopathy or cardiac phenotypes (e.g., dilated cardiomyopathy or Mobitz type II AV block) indicating DFNA10. One patient underwent unilateral CI at the age of 80, due to gradual hearing deterioration. Following CI, her sentence recognition score (K-CID) improved significantly, from 18% at baseline to 60% at 2 years postoperatively (Figure 2d).

## 4. Discussion

This study is the first to provide detailed genotype and audiological phenotypes associated with TF variants inducing non-syndromic deafness. In the clinical exome sequencing era, many questions regarding the pathogenic mechanisms of SNHL have been answered, allowing a functional classification of the etiology of genetic hearing loss. Based on our in-house databases of genetic hearing loss, TF genes were implicated in ~3% of the study patients. Notably, 33 potentially pathogenic variants were observed, including nine novel variants, accounting for non-syndromic deafness clustered in only four TF genes (*POU3F4*, *POU4F3*, *LMX1A*, and *EYA4*), indicating a narrow molecular etiologica; spectrum within the enormous number of TF genes reported thus far in humans (up to 1600 genes). The limited genetic spectrum of TF genes accounting for non-syndromic deafness suggests the functional redundancy of many other TF genes in inner ear development or the maintenance of function. Alternatively, fetuses with variants in developmentally lethal, deafness-related TF genes may be spontaneously aborted.

*POU3F4* encodes a POU-domain TF expressed in mesenchymal cells of the otic capsule, responsible for normal inner ear development [21,22]. *POU3F4* mutant mice have reduced endocochlear potential and alterations in cochlear spiral ligament fibrocytes [23], leading to hearing impairment. *POU3F4* is associated with IP type III, segregating as an X-linked recessive trait. In our cohort, *POU3F4* was the TF gene most commonly associated with non-syndromic deafness. Although in most patients, single-nucleotide variants at the *POU3F4* locus were identified, structural variants such as a large deletion upstream of the *POU3F4* locus [24,25,26], have been reported. The current study also expands the genotypic spectrum of disease-causing variants of *POU3F4* causing DFNX2, including the detection of four novel variants.

Despite numerous reports on CI in patients with *POU3F4* variants, detailed information is lacking and the postoperative outcomes have been highly variable. Choi et al. [25] and Smeds et al. [22] reported that patients with *POU3F4* variants had improved speech scores after implantation but the scores were still lower than those of age-matched cohorts without apparent cochlear anomalies. These results differ from those reported by Tian et al. [27] and Kang et al. [28]. To our knowledge, the present study was based on the largest cohort of patients and families with genetically confirmed IP type III, with 11 patients (13 ears) undergoing CI. The postoperative speech performance of *POU3F4* patients was notably poorer than that of the *GJB2*-control group, precluding a potential correlation between postoperative CI outcomes and genotype. None of the 13 implanted ears of our *POU3F4* cohort exhibited either a change or deterioration in speech or aided hearing upon follow-up beyond 3 years. Either intraoperative CSF gushers during electrode insertion or an unexpected localization of the spiral ganglion neurons might hinder the CI outcome, even in a patient with a genotype allowing residual transcriptional performance. Chao et al. [29] also reported an inconsistent distribution and responsiveness of the residual spiral ganglion neuron in IP type III, with CI outcomes varying accordingly. Furthermore, a recent study showed that *POU3F4* variants were associated with neurodevelopmental disorders, such as hyperactivity, concentration difficulties, poor phonological working memory, and slow language development [22], all of which may contribute to a negative CI outcome.

*POU4F3*, a member of the POU family of TFs, is encoded by a gene located on chromosome 5q32 [30] and comprises two highly conserved POU domains: a POU-specific domain and a POU homeodomain [31]. *POU4F3* is expressed in cochlear hair cells and plays a pivotal role in their differentiation, maturation, and maintenance by regulating downstream transcripts [32]. In humans, *POU4F3* defects commonly cause autosomal dominant deafness, with variants associated with progressive non-syndromic deafness of postlingual onset [33,34,35,36,37,38,39]. The results of this study suggest that *POU4F3* is the most common TF gene to cause non-syndromic deafness, excluding *POU3F4* associated with X-linked inherited deafness in IP type III. In terms of related audiological characteristics, Kitano et al. [38] reported that *POU4F3*-associated DFNA15 is primarily characterized by mid-frequency hearing loss, followed by high-frequency hearing loss. This is consistent with our findings, as 45.5% of our cohort had U-shaped audiograms (mid-frequency hearing loss), followed by 36.4% with down-sloping audiograms (high-frequency hearing loss). Unfortunately, few studies have examined CI outcomes in patients with the *POU4F3* variant. Kitano et al. [38] reported on two individuals with *POU4F3* deafness who underwent CI, with good postoperative auditory performance. Miyake et al. [40] also reported a favorable CI outcome in a patient with a *POU4F3* mutation. Three of our CI recipients with *POU4F3* variants had a favorable outcome even at 1 year postoperatively. Our study thus doubles the number of reported CIs in patients with *POU4F3* variants. In line with the spiral ganglion neuron hypothesis [41], these results suggest that *POU4F3* patients are promising candidates for CI, as favorable postoperative outcomes can be expected.

*LMX1A*, a LIM homeobox TF, has been recently implicated in non-syndromic deafness. *LMX1A* plays a vital role in ear patterning, regulating the morphogenesis of inner ear structures [18]. To date, only nine *LMX1A* variants have been reported in the literature. Wesdorp et al. [42] first reported two Dutch families with *LMX1A* dominant variants (c.290G>C;p.Cys97Ser and c.721G>C;p.Val241Leu) and variable onset, severity, progression, and asymmetry. Schrauwen et al. [43] described a Pakistani family with an *LMX1A* recessive C-terminus missense variant (c.1106T>C;p.Ile369Thr) associated with profound SNHL. In recent studies, we reported five *LMX1A* heterozygous variants (c.595A>G:p.Arg199Gly, c.622C>T;p.Arg208*, c.719A>G;p.Gln240Arg, c.721G>A;p.Val241Met, and c.887dup;p.Gln297Thrfs*41) in five Korean families and one *LMX1A* heterozygous variant (c.686C>A; p.Ala229Asp) in a Polish family [18,44]. Together, these results suggest that alterations in *LMX1A* are associated with dominantly inherited asymmetric deafness. In these variants, audiological severity correlated with the extent of transcriptional activation, measured in a luciferase-reporter assay, implying a genotype–phenotype correlation [18]. Our study expands the genotypic spectrum of *LMX1A* in DFNA7 by identifying a novel variant (c.331del:p.Gln111Argfs*7), in addition to demonstrating audiologic asymmetry (SH421). Indeed, most *LMX1A*-related DFNA7 patients display audiologic asymmetry to varying degrees and a gradual progression of hearing loss. Our results provide evidence for favorable CI outcomes in patients with *LMX1A*-related DFNA7.

*EYA4* encodes both an EYA domain (eya-HR) and a transactivation domain (eya-VR) and is responsible for the function of the mature organ of Corti [45]. *EYA4* mutations generally cause autosomal dominant non-syndromic deafness (DFNA10). However, in some cases, *EYA4* variants result in cardiac phenotypes, including dilated cardiomyopathy and a Mobitz type II AV block with deafness [46,47]. Makishima et al. [46] and Schonberger et al. [48] proposed that the variant location of *EYA4* determines the occurrence of cardiac abnormalities, and thus a genotype–phenotype correlation. Truncating variants residing in the eya-HR domain seem to be associated with non-syndromic deafness, whereas those residing in the eya-VR region lead to dilated cardiomyopathy with deafness. However, recent studies have reported additional cases implicating truncations in the eya-VR region with non-syndromic deafness [49,50]. Three of our five *EYA4* variants were predicted to encode truncated proteins affecting the eya-VR domain, but cardiac evaluations in these patients showed no signs of dilated cardiomyopathy. Our study also expands the genotypic spectrum of *EYA4* in DFNA10, by adding four novel variants (c.697C>T:p.Gln233Ter, c.208+1del, c.578dup:p.Tyr193Ter, and c.1468G>T:p.Glu490Ter), all of which were associated with moderate to severe non-syndromic deafness characterized by gradual hearing loss. To our knowledge, this is the first study to describe a successful CI outcome in a patient with an *EYA4* variant, but this result must be confirmed in a larger cohort.

Although in this study we examined the clinical characteristics of disease-causing TF variants associated with non-syndromic deafness, the study’s limitations should be addressed in future studies. A larger cohort with longitudinal audiologic follow-up is needed to strengthen our findings. Furthermore, functional studies of different genotypes associated with TF genes, such as transcriptional activity, nuclear localization, and mutant protein stability, will reveal the clinical and molecular relationships of TF genes. Nevertheless, our results provide further insights into the genetic landscape of TF-related non-syndromic deafness and thus a basis for the implementation of a personalized, genetically tailored approach for audiological treatment and rehabilitation in these patients.

## Figures and Tables

**Figure 1 biomedicines-10-02125-f001:**
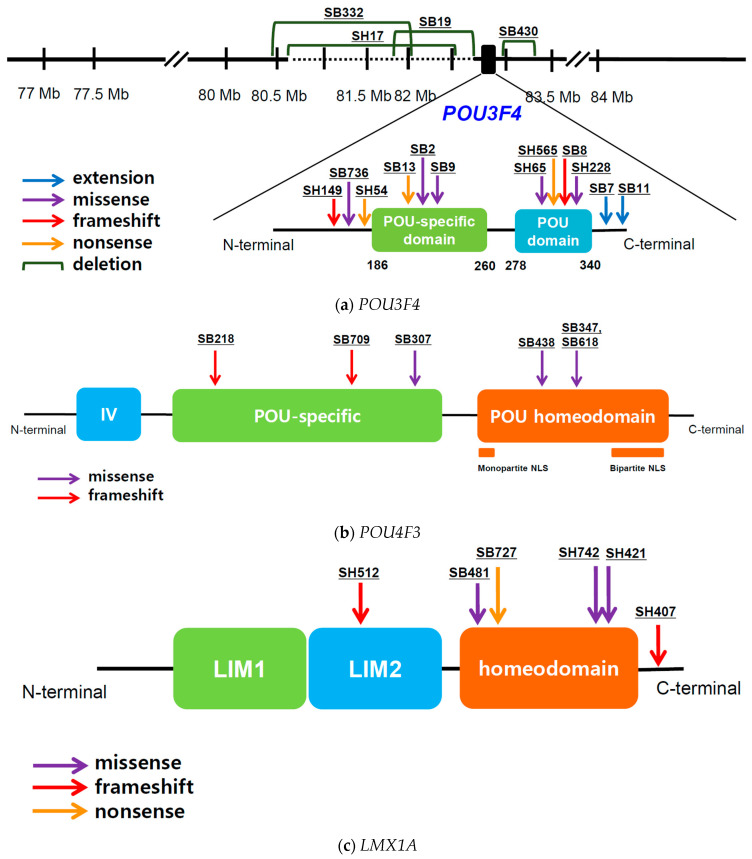
Physical map of variants caused by four transcription factor genes associated with non-syndromic deafness: (**a**) *POU3F4*; (**b**) *POU4F3*; (**c**) *LMX1A*; (**d**) *EYA4*.

**Figure 2 biomedicines-10-02125-f002:**
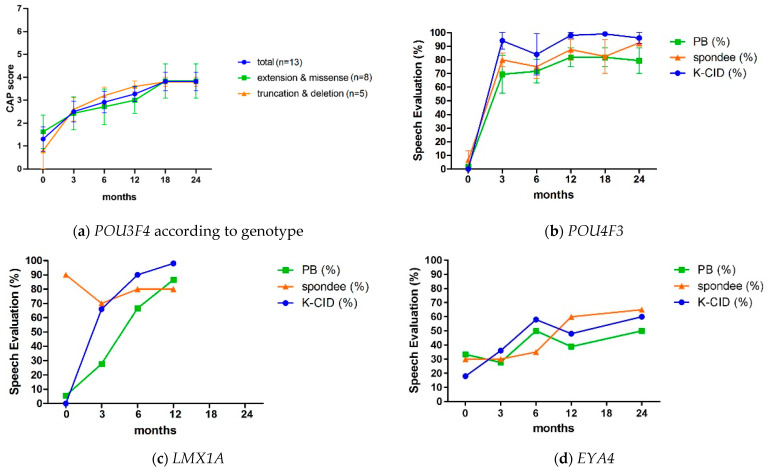
Postoperative cochlear implant outcomes in patients with mutations in transcription factor genes: (**a**) *POU3F4* according to genotype; (**b**) *POU4F3*; (**c**) *LMX1A*; (**d**) *EYA4*.

**Table 1 biomedicines-10-02125-t001:** Phenotypes and genotypes associated with non-syndromic deafness caused by transcription factor variants.

Patient	Sex	Timing of HL	Genotype	Age at HL Detection	Type of HL	Audiogram Configuration	Degree of HL (Most Recent)	Asymmetry	HL Progression	Final Aural Rehabilitation	Age at CI	Allele Frequency ^e^
SB2-1	M	prelingual	*POU3F4*[NM_000307.4]c.626A>G:p.Gln229Arg	1 month	SNHL	Flat	profound	No	No	B) CI	R) 2 yr, L) 12 mo	absent
SB2-2	M	prelingual	*POU3F4*[NM_000307.4]c.626A>G:p.Gln229Arg	6 months	SNHL	Flat	profound	No	No	B) CI	R) 6 yr, L) 7 yr	absent
SB7	M	prelingual	*POU3F4*[NM_000307.4]c.1060delA:p.Thr354Glnfs*115	12 months	SNHL	Flat	profound	No	No	CI	2 yr	absent
SB8	M	postlingual	*POU3F4*[NM_000307.4]c.950dupT:p.Leu317Phefs*12	35 months	MHL	Mixed HL	severe	No	Yes^d^ (2.25 dB HL/yr)	HA	(-)	absent
SB9	M	postlingual	*POU3F4*[NM_000307.4]c.632C>T:p.Thr211Met	3 years	MHL	Mixed HL	severe	Yes (24 dB)	Yes (0.7 dB HL/yr)	HA	(-)	absent
SB11	M	prelingual	*POU3F4*[NM_000307.4]c.1084T>C:p.X362Argext*113	3 years	SNHL	Flat	profound	No	No	CI	12 yr	absent
SB13	M	prelingual	*POU3F4*[NM_000307.4]c.623T>A:p.Leu208*	15 months	SNHL	Flat	profound	No	Yes (0.8 dB HL/yr)	B) CI	R) 6 yr, L) 2yr	absent
SH17	M	prelingual	Xq21.2, 80851535-82597832 bp	1 month	SNHL	Flat	profound	No	Yes (4.7 dB HL/yr)	B) CI	R) 13 mo, L) 25 mo	absent
SB19	M	prelingual	Xq21.2, 81810457-82810060 bp	14 months	SNHL	Flat	profound	No	No	CI	29 yr	absent
SH54	M	postlingual	*POU3F4*[NM_000307.4]c.540C>A:p.Cys180*	1 month	MHL	Mixed HL	severe	No	Yes (1.4 dB HL/yr)	HA	(-)	absent
SH65	M	prelingual	*POU3F4*[NM_000307.4]c.910C>A:p.Pro303His	1 month	SNHL	Downsloping	severe	Yes (15 dB)	Yes (0.5 dB HL/yr)	CI	3 yr	absent
SH149	M	prelingual	*POU3F4*[NM_000307.4]c.458delC:p.Pro153Leufs*88	3 months	MHL	Mixed HL	profound	Yes (18 dB)	Yes (2.5 dB HL/yr)	CI	3 yr	absent
SH228	M	prelingual	*POU3F4*[NM_000307.4]c.989G>A:p.Arg330Lys	unknown ^a^	MHL	Mixed HL	severe	No	No	HA	(-)	absent
SB332	M	prelingual	Xq21.2, deletion	1 month	MHL	N/A ^b^	severe	No	No	HA	(-)	absent
SB430	M	prelingual	Xq21.2, deletion	1 month	MHL	N/A ^b^	severe	No	No	CI	21 mo	absent
SH565-1	M	prelingual	*POU3F4*[NM_000307.4]c.958G>T:p.Glu320*	1 month	MHL	Mixed HL	moderate	No	No	HA	(-)	absent
SH565-2	M	prelingual	*POU3F4*[NM_000307.4]c.958G>T:p.Glu320*	2 month	MHL	Mixed HL	moderately severe	No	No	HA	(-)	absent
SB736	M	prelingual	*POU3F4*[NM_000307.4]c.626A>G:p.Gln229Arg	12 months	MHL	Mixed HL	profound	No	No	B) CI ^c^	10 yr	absent
SB218	F	postlingual	*POU4F3*[NM_002700.2] c.564dupA:p.Ala189Serfs*26	30 years	SNHL	U-shaped	moderate	No	Yes (1.6 dB HL/yr)	B) MEI	(-)	absent
SB307	F	postlingual	*POU4F3*[NM_002700.2] c.743T>C:p.Leu248Pro	26 years	SNHL	U-shaped	moderately severe	No	Yes	HA	(-)	absent
SB347-1	F	postlingual	*POU4F3*[NM_002700.2] c.952G>A:p.Val318Met	16 years	MHL	Mixed HL	profound	No	Yes (16.7 dB HL/yr)	CI	36 yr	absent
SB347-2	F	postlingual	*POU4F3*[NM_002700.2] c.952G>A:p.Val318Met	20 years	SNHL	Flat	profound	No	Yes	CI	52 yr	absent
SB438-1	F	postlingual	*POU4F3*[NM_002700.2] c.879C>A:p.Phe293Leu	unknown ^a^	SNHL	Downsloping	mild	No	Yes	HA	(-)	absent
SB438-2	F	postlingual	*POU4F3*[NM_002700.2] c.879C>A:p.Phe293Leu	37 years	SNHL	Downsloping	moderately severe	No	Yes (2.3 dB HL/yr)	HA	(-)	absent
SB618-1	M	prelingual	*POU4F3*[NM_002700.2] c.952G>A:p.Val318Met	unknown ^a^	SNHL	U-shaped	moderate	No	Yes (5 dB HL/yr)	HA	(-)	absent
SB618-2	M	unknown ^a^	*POU4F3*[NM_002700.2] c.952G>A:p.Val318Met	unknown ^a^	SNHL	U-shaped	severe	Yes (16 dB)	unknown ^a^	HA	(-)	absent
SB618-3	F	unknown ^a^	*POU4F3*[NM_002700.2] c.952G>A:p.Val318Met	unknown ^a^	SNHL	Downsloping	R) severe, L) profound	Yes (44 dB)	unknown ^a^	HA	(-)	absent
SB709	F	postlingual	*POU4F3*[NM_002700.2] c.662_675del:p.Gly221Glufs*77	39 years	SNHL	U-shaped	profound	No	Yes (10.6 dB HL/yr)	B) CI ^c^	36 yr	absent
SB481	M	prelingual	*LMX1A*[NM_177398.4] c.595A>G:p.Arg199Gly	1 months	SNHL	N/A ^b^	R) profound, L) severe	Yes (15 dB)	unknown ^a^	HA	(-)	absent
SB727	F	postlingual	*LMX1A*[NM_177398.4] c.622C>T:p.Arg208*	13 years	SNHL	Downsloping	R) moderate, L) profound	Yes (36 dB)	Yes	CI	32 yr	absent
SH407	F	postlingual	*LMX1A*[NM_177398.4] c.887dup:p.Gln297Thrfs*41	20 years	SNHL	Downsloping	R) moderately severe, L) moderate	Yes (18 dB)	fluctuation	HA	(-)	absent
SB742-1	F	prelingual	*LMX1A*[NM_177398.4] c.719A>G:p.Gln240Arg	2 months	SNHL	N/A ^b^	R) moderate, L) profound	Yes (45 dB)	No	HA	(-)	absent
SB742-2	F	postlingual	*LMX1A*[NM_177398.4] c.719A>G:p.Gln240Arg	20 years	SNHL	Downsloping	moderately severe	No	Yes	(-)	(-)	absent
SH421-1	F	prelingual	*LMX1A*[NM_177398.4] c.721G>A:p.Val241Met	4 months	SNHL	N/A ^b^	moderate	No	unknown ^a^	HA	(-)	absent
SH421-2	M	postlingual	*LMX1A*[NM_177398.4] c.721G>A:p.Val241Met	17 years	SNHL	Downsloping	R) profound, L) moderate	Yes (61 dB)	No	HA	(-)	absent
SH512-1	F	prelingual	*LMX1A*[NM_177398.4] c.331del:p.Gln111Argfs*7	3 months	SNHL	N/A ^b^	moderate	No	unknown ^a^	(-)	(-)	absent
SH512-2	F	prelingual	*LMX1A*[NM_177398.4] c.331del:p.Gln111Argfs*7	1 year	SNHL	Downsloping	severe	No	unknown ^a^	HA	(-)	absent
SB302-1	F	postlingual	*EYA4*[NM_004100.5] c.697C>T:p.Gln233*	35 years	SNHL	U-shaped	moderate	No	Yes	(-)	(-)	1/3444 (KRGDB)
SB302-2	F	postlingual	*EYA4*[NM_004100.5] c.697C>T:p.Gln233*	40 years	SNHL	Flat	severe	No	Yes	HA	(-)	1/3444 (KRGDB)
SB545	F	postlingual	*EYA4*[NM_004100.5] c.208+1del	50 years	MHL	Downsloping	severe	No	Yes	CI	80 yr	absent
SB865	F	postlingual	*EYA4*[NM_004100.5] c.578dup:p.Tyr193*	10 years	MHL	Downsloping	severe	Yes (54 dB)	No	HA	(-)	absent
SH537	F	postlingual	*EYA4*[NM_004100.5] c.1468G>T:p.Glu490*	45 years	SNHL	Flat	moderately severe	No	Yes	HA	(-)	1/3444 (KRGDB)
SH117-1	F	postlingual	*EYA4*[NM_004100.5] c.1194del:p.Met401Trpfs*3	15 years	SNHL	Downsloping	moderate	No	Yes	HA	(-)	absent
SH117-2	M	postlingual	*EYA4*[NM_004100.5] c.1194del:p.Met401Trpfs*3	unknown ^a^	SNHL	Downsloping	severe	No	unknown ^a^	(-)	(-)	absent
SH117-3	F	postlingual	*EYA4*[NM_004100.5] c.1194del:p.Met401Trpfs*3	unknown ^a^	SNHL	Downsloping	moderate	No	unknown ^a^	(-)	(-)	absent

M = male, F = female, B) = both, R) = right, L) = left, HL = hearing loss, dB = decibel, SNHL = sensorineural heaing loss, MHL = mixed hearing loss, HA = hearing aid, CI = cochlear implant, MEI = middle ear implant, yr = year, mo = month. ^a^ Unknown, due to lack of record. ^b^ Non-applicable, because the patient was too young to undergo pure tone audiometry. ^c^ Cochlear implant was done on both ears simultaneously. ^d^ Hearing loss progression was observed in bone conduction only. ^e^ Allele frequency in Korean control population, from KRGDB (Korean Reference Genome Database, http://152.99.75.168:9090/KRGDB, accessed on 17 August 2022) [20].

**Table 2 biomedicines-10-02125-t002:** Novel variants of transcription factor genes associated with non-syndromic deafness.

Patient	Genotype	dbSNP ID (dbSNP v151)	Zygosity	Inheritance	ACMG Guideline	
					Classification	Criteria
SH149	*POU3F4*[NM_000307.5]c.458delC:p.Pro153Leufs*88	absent	hemizygote	XLR	Likely pathogenic	PVS1, PM2, PP4
SH228	*POU3F4*[NM_000307.5]c.989G>A:p.Arg330Lys	absent	hemizygote	XLR	Likely pathogenic	PS2_moderate, PM2, PP3, PP4
SH565	*POU3F4*[NM_000307.5]c.958G>T:p.Glu320*	absent	hemizygote	XLR	Likely pathogenic	PVS1, PM2, PP4
SB736	*POU3F4*[NM_000307.5]c.499C>T:p.Arg167*	rs111033345	hemizygote	XLR	Pathogenic	PVS1, PM2, PP1_strong, PM3_strong, PP4
SH512	*LMX1A*[NM_177398.4] c.331del:p.Gln111Argfs*7	absent	heterozygote	AD	Likely pathogenic	PVS1, PM2
SB302	*EYA4*[NM_004100.5] c.697C>T:p.Gln233*	rs1583346685	heterozygote	AD	Likely pathogenic	PVS1, PM2
SB545	*EYA4*[NM_004100.5] c.208+1del	absent	heterozygote	AD	Likely pathogenic	PVS1, PM2, PP3
SB865	*EYA4*[NM_004100.5] c.578dup:p.Tyr193*	absent	heterozygote	AD	Likely pathogenic	PVS1, PM2, PP1_mod
SH537	*EYA4*[NM_004100.5] c.1468G>T:p.Glu490*	rs1305000119	heterozygote	AD	Likely pathogenic	PVS1, PM2

dbSNP = Single Nucleotide Polymorphism Database, ACMG = the American College of Medical Genetics and Genomics, XLR = X-linked recessive, AD = autosomal dominant, PVS1 = very strong evidence of pathogenicity, PS = strong evidence of pathogenicity, PM = moderate evidence of pathogenicity, PP = supporting evidence of pathogenicity.

**Table 3 biomedicines-10-02125-t003:** Postoperative auditory perception score (CAP score) of cochlear implantees with *POU3F4* variants.

Cochlear Implantees	Preoperative	Post-CI 3 Months	Post-CI 6 Months	Post-CI 12 Months	Post-CI 18 Months	Post-CI 24 Months
*POU3F4* (*n* = 13)	1.3	2.6	3.0	3.4 *	3.7 *	3.8 *
*GJB2* control ** (*n* = 26)	1.1	2.2	3.8	5.8	6.2	6.7

CI = cochlear implant. * The difference was statistically significant compared to the control group. (*p* < 0.05) ** age-, sex-, laterality-matched control group of *GJB2*-associated cochlear implantees.

## Data Availability

The data presented in this study are available on request from the corresponding author upon reasonable request.

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
