# Peer review of "Genetic Load of Alternations of Transcription Factor Genes in Non-Syndromic Deafness and the Associated Clinical Phenotypes: Experience from Two Tertiary Referral Centers"

_biomedicines, 2022, doi:10.3390/biomedicines10092125_

Round 1
Reviewer 1 Report
Jo and colleagues collected and analyzed the genomic results of 1,280 probands with hearing impairment. Among the 720 probands whose disease-causing variants were confirmed, the authors selected 33 study probands with causative variants in four well-known transcription factor (TF) genes, i.e., POU3F4, POU4F3, LMX1A, and EYA4, for further investigation. A total of nine novel deafness-causing variants in the four TF genes were identified. The authors then analyzed and characterized the audiological features and cochlear implantation outcomes in the 33 subjects.
In general, this is a well-organized and well-written article. A major strength of this article lies in the expansion of the genotypic spectrum of disease-causing variants in the four TF genes and the characterization of the auditory phenotypes associated with the genotypes. The reviewer thinks that the content of this article is of high clinical value in genetic diagnosis for hearing loss, particularly in the era with rapid growth and accumulation of high-throughput genomic data. This paper would be better off by addressing the following points.
Comments:
1. What are the allele frequencies of these TF gene variants identified in this study in the Korean population?
2. I just noticed an interesting case in Table 1. SB218, who is a POU4F3 case with bilateral moderate SNHL, underwent bilateral middle ear implant surgery. The authors may want to elaborate more on this case.
Author Response
[Reviewer 1]
1. What are the allele frequencies of these TF gene variants identified in this study in the Korean population?
-> We added a column titled "allele frequency" in [Table 1], describing the allele frequency in Korean control population for each TF gene variants. The data was extracted from the Korean Reference Genome Database.
2. I just noticed an interesting case in Table 1. SB218, who is a POU4F3 case with bilateral moderate SNHL, underwent bilateral middle ear implant surgery. The authors may want to elaborate more on this case.
-> We elaborated on SB218's history of hearing loss and aural rehabilitation, by adding the following sentence at the end of the subsection [3.3.2. POU4F3: audiological profile and cochlear implantation results]: "One patient (SB216) displayed bilateral moderate SNHL in her early 30's and opted for bilateral middle ear implantation (MEI) surgery rather than hearing aid due to unsatisfactory experience with conventional hearing aids. She has been a satisfied user of MEI for 6 years."
Reviewer 2 Report
The information is important to a certain audience, I am sure, but the writing style is not the most engaging. A hook, to draw the reader in and want to read more would be an improvement.
Author Response
Thank you very much for your suggestion. We have tried our best to draw readers' attention further in the revised version.